# MicroRNA Signatures Associated with Bronchopulmonary Dysplasia Severity in Tracheal Aspirates of Preterm Infants

**DOI:** 10.3390/biomedicines9030257

**Published:** 2021-03-05

**Authors:** Roopa Siddaiah, Christiana N. Oji-Mmuo, Deborah T. Montes, Nathalie Fuentes, Debra Spear, Ann Donnelly, Patricia Silveyra

**Affiliations:** 1Department of Pediatrics, The Pennsylvania State University College of Medicine, Hershey, PA 17033, USA; rsiddaiah@pennstatehealth.psu.edu (R.S.); cojimmuo@pennstatehealth.psu.edu (C.N.O.-M.); dspear@pennstatehealth.psu.edu (D.S.); adonnelly@pennstatehealth.psu.edu (A.D.); 2Biobehavioral Laboratory, School of Nursing, The University of North Carolina at Chapel Hill, Chapel Hill, NC 27599, USA; dmontes@email.unc.edu; 3National Institute of Allergy and Infectious Diseases, National Institutes of Health, Bethesda, MD 20892, USA; nathalie.fuentesortiz@nih.gov; 4Department of Environmental and Occupational Health, School of Public Health, Indiana University, Bloomington, IN 47405, USA

**Keywords:** bronchopulmonary dysplasia, prematurity, miRNA, biomarkers, tracheal aspirates

## Abstract

Bronchopulmonary dysplasia (BPD) is a form of chronic lung disease that develops in neonates as a consequence of preterm birth, arrested fetal lung development, and inflammation. The incidence of BPD remains on the rise as a result of increasing survival of extremely preterm infants. Severe BPD contributes to significant health care costs and is associated with prolonged hospitalizations, respiratory infections, and neurodevelopmental deficits. In this study, we aimed to detect novel biomarkers of BPD severity. We collected tracheal aspirates (TAs) from preterm babies with mild/moderate (*n* = 8) and severe (*n* = 17) BPD, and we profiled the expression of 1048 miRNAs using a PCR array. Associations with biological pathways were determined with the Ingenuity Pathway Analysis (IPA) software. We found 31 miRNAs differentially expressed between the two disease groups (2-fold change, false discovery rate (FDR) < 0.05). Of these, 4 miRNAs displayed significantly higher expression levels, and 27 miRNAs had significantly lower expression levels in the severe BPD group when compared to the mild/moderate BPD group. IPA identified cell signaling and inflammation pathways associated with miRNA signatures. We conclude that TAs of extremely premature infants contain miRNA signatures associated with severe BPD. These may serve as potential biomarkers of disease severity in infants with BPD.

## 1. Introduction

Bronchopulmonary dysplasia (BPD) is a form of chronic lung disease that develops in neonates as a consequence of preterm birth and arrested fetal lung development [1]. BPD is associated with significant human and public health burdens, with reported cases varying significantly among centers worldwide [2]. The incidence of BPD remains on the rise, with recently stated global incidence ranging from 10% to up to 89% in one report [2,3]. The disease was first described by Northway and colleagues in 1967 in preterm infants receiving invasive mechanical ventilation with characteristic clinical, chest radiological, and pathologic findings [4]. 

The etiology and pathogenesis of BPD are complex and multifactorial. The contributing factors include genetic predisposition, epigenetic factors, arrest of lung development, chronic inflammation, mechanical ventilation, and oxygen toxicity [1,5]. In addition, several maternal health factors such as pre-eclampsia, obesity, gestational diabetes, and inflammation can also predispose to preterm birth and BPD development [6,7,8]. Severe BPD is particularly prevalent in extremely low birth weight infants, a fast-growing population. Fetal growth restriction and male gender are additional risk factors for severe BPD [5,9]. Neonates with severe BPD are at risk for adverse short- and long-term outcomes that are potentially life-long [10]. Preterm infants who are survivors of BPD also experience prolonged hospitalizations, increased incidence of respiratory infections, growth failure, and neurodevelopmental deficits [10].

Clinically, BPD presents as chronic respiratory insufficiency, with oxygen requirement with or without the need for positive pressure support [1]. Historically, the incidence and classification of BPD have evolved due to the improved survival of extremely low birth weight infants, advances in neonatal intensive care, the evolution in the modes of mechanical ventilation, and other therapies such as exogenous surfactant and corticosteroids [11,12]. While BPD was previously classified by the National Institute of Child Health and Human Development (NICHD) into three categories (mild, moderate, or severe), since 2019 the same committee began to use a staging system for the classification of BPD, with grades I, II, and III to reflect mild, moderate, and severe disease, respectively [13,14]. As opposed to the initial classification system that focused on chronicity and extent of respiratory support, the more recent classification attempts to classify BPD severity based on short- and long-term outcomes such as death, need for tracheostomy, and hospital readmission rates. While these systems incorporate clinical phenotyping into diagnosis, we consider that synergizing them with endotyping through omics markers should be the future direction.

In the past decade, studies have proliferated looking at different genomic, transcriptomic, epigenomic, metabolomics, and other biological biomarkers for the prediction of BPD and its severity [15,16,17,18]. Animal models of newborn hyperoxia have identified several pathways involved in the development of BPD, including the cyclooxygenase-2 (COX-2) [19,20], sirtuin-1 (SIRT1) [21], and Wnt signaling pathways [22]. To date, there are no validated biomarkers for predicting risk and disease severity in the neonatal period. The current consensus is that biomarker development could lead to early recognition, individualized care, and interventions that could prevent severe BPD in the most vulnerable group, subsequently improving short- and long-term outcomes. Particular biomarkers of interest examined in the present study are microRNAs (miRNAs) from tracheal aspirates (TAs). miRNAs are small noncoding molecules that are approximately 22 nucleotides long. They regulate gene expression post-transcriptionally by inhibiting the expression of target messenger RNAs [23]. A few individual miRNAs have been identified as key players of BPD pathogenesis in mouse models and clinical studies. A recent study identified miR-219-5p as a marker for severe BPD in preterm infants when compared to full-term neonates [24]. Another study using cord blood and venous blood from infants with BPD identified an association of decreased expression of miR-29 and intra-amniotic and elevated inflammatory markers [25]. 

The potential of TA miRNAs to predict disease severity in neonatal and pediatric lung disease is promising and warrants additional investigation. This is highlighted by the fact that TAs are obtained relatively easily from endotracheal tube suctioning in infants receiving invasive mechanical ventilation [26,27]. Moreover, TAs are a vital source of information to study molecular pathways related to the microenvironment of the developing lung and could not only provide a source of novel biomarkers for BPD diagnosis but also identify molecular events occurring during lung development in the context of the disease [26,28,29]. 

With this goal in mind, the aim of the present study was to analyze the expression profile of miRNAs present in TAs obtained from a cohort of neonates with mild/moderate and severe BPD. This study was conducted as part of a larger single-center prospective cohort study with some of the results reported by us previously [18]. While our prior analysis identified miRNA signatures of prematurity and BPD pathogenesis in TAs, the current study focuses on specific miRNAs associated with BPD severity, in order to help develop more precise diagnostic markers.

## 2. Materials and Methods

### 2.1. Patient Population

We prospectively enrolled 25 extremely preterm infants receiving invasive mechanical ventilation at the Penn State Health Children’s Hospital Neonatal Intensive Care Unit between 2013 and 2018. Enrolled patients were 7 days of life or older and at risk for developing mild, moderate, or severe BPD according to the NHLBI consensus conference classification [30]. In short, the severity of BPD was determined based on the degree of respiratory support and/or oxygen requirement at 28 days of life and 36 weeks postmenstrual age or at 56 days of life, whichever was earlier. Infants with major congenital malformations, chromosomal anomalies, and complex congenital heart defects were excluded from the study [31]. We also excluded infants with severe lung disease and signs of pulmonary hypertension in echocardiogram at or close to 36 weeks of GA. All protocols used in this study were approved by the Pennsylvania State University College of Medicine Institutional Review Board on 5 March 2013 under protocol ID PRAMS042136EP. 

### 2.2. Tracheal Aspirate Collection

Following informed consent by the parents, we collected TAs from enrolled patients following routine suctioning. We also obtained pertinent clinical information from electronic medical records, such as sex, race/ethnicity, method of delivery, antenatal steroids, gestational age (GA) at birth, birth weight, GA at the time of sample collection, day of life, and fraction of inspired oxygen (FiO_2_) at the time of sample collection. Immediately after collection, TAs were transported to the laboratory and stored at −80 °C until analysis. 

### 2.3. MicroRNA Purification

Starting from 500 µL of TAs, we purified miRNAs using the Norgen miRNA Purification Kit, after addition of a spike-in control (cel-mir-39, QIAGEN Inc., Germantown, MD, USA). The miRNA fraction was then quantified using a Nanodrop, and its purity was assessed with a Bioanalyzer at the Penn State College of Medicine Genome Sciences Core Facility. For miRNA profiling, a total of 250 ng of RNA was first retrotranscribed using the miScriptII RT kit (QIAGEN Inc., Germantown, MD, USA), following the manufacturer’s protocol. 

### 2.4. MicroRNA Arrays

The expression of 1066 human miRNAs was quantified using the miScript miRNA PCR array kit MIHS-3216Z (QIAGEN Inc., Germantown, MD, USA) on a QuantStudio 12K Flex system (Life Technologies, Carlsbad, CA, USA). Following the PCR reaction, cycle threshold (Ct) values were extracted and normalized by global means. Differential expression was calculated using the 2^−ΔΔCT^ method and a mild/moderate sample as a calibrator [32]. Fold changes in miRNA expression were obtained using the Bioconductor *limma* package on R [33]. miRNAs with undetectable expression (Ct values > 40) in more than 90% of samples of each experimental group were excluded from the analysis. Datasets (metadata, non-normalized, and normalized values) were uploaded to the Gene Expression Omnibus under project GSE165828. 

### 2.5. Ingenuity Pathway Analysis

The Ingenuity Pathway Analysis (IPA) software (QIAGEN Inc., Germantown, MD, USA) was used to identify miRNA interaction networks, predicted target genes, and molecular functions based on prediction scores [34]. The IPA core analysis functionality was performed for both direct and indirect relationships using the Ingenuity Knowledge Base. The miRNAs of interest were analyzed using the microRNA target filter feature. Only experimentally observed or predicted with high confidence targets in humans were considered.

### 2.6. MicroRNA Expression Validation

The expression of six differentially expressed miRNAs between mild/moderate and severe BPD samples was validated using the All-in-One miRNA qRT-PCR Reagent Kits (GeneCopoeia Inc., Rockville, MD, USA), using selected primers (hsa-miR-205-3p, hsa-let-7i-5p, hsa-miR-1255b, hsa-miR-24-1-5p, hsa-miR-545-5p, and hsa-miR-628-5p) and starting from 6 ng of RNA template. Differential expression between groups was calculated with the 2^−ΔΔ*C*T^ method using hsa-miR-16-1-3p as a normalization control [32,35].

### 2.7. Statistical Analysis and Code Availability

For miRNA arrays, statistical analyses were performed using the Bioconductor *limma* package on R using the parametric empirical Bayes method [33]. Differential expression was defined as Benjamini–Hochberg false discovery rate (FDR) below 0.05. Heatmaps were generated with the nonnegative matrix factorization (NMF) package on R [36], and volcano plots were generated using the GraphPad Prism software. For miRNA validation assays, differences in miRNA expression between the two groups were determined by *t*-test using the GraphPad Prism software and considered statistically significant at *p* < 0.05.

## 3. Results

### 3.1. Patient Demographics 

We obtained TAs from 25 mechanically ventilated preterm infants with mild/moderate BPD (*n* = 8) and severe BPD (*n* = 17). A summary of the patients’ clinical information is shown in Table 1, and individual data for main variables are visualized in Figure 1. As expected, FiO2 at the time of sample collection was significantly higher in the severe BPD group when compared to the mild/moderate BPD group (*p* < 0.05). No statistically significant differences were found for the rest of the measured variables, including GA at birth and at sample collection time, birth weight, day of life at sample collection time, male sex, and antenatal steroid exposure. Regarding the racial and ethnic composition of the patient cohort, there was a significant overrepresentation of Hispanic and mixed-race subjects in the severe BPD group when compared to the mild/moderate BPD group (*p* < 0.05).

### 3.2. MiRNA Expression in TAs

A heatmap of the expression of the 1048 miRNAs expressed in TA samples from mild/moderate and severe BPD infants is shown in Figure 2. A volcano plot indicating FDR vs. fold change (FC) of expression between severe BPD and mild/moderate BPD samples is also shown in Figure 3. 

After accounting for FDR < 0.05, the PCR array analysis revealed differential expression of 31 miRNAs with |FC| > 2 between the mild/moderate and severe BPD samples (Table 2). Of these, 4 miRNAs had significantly higher expression in the severe BPD group (log FC > 0.3, FDR < 0.05) and 27 had significantly higher expression in the mild/moderate BPD group (log FC < −0.3, FDR < 0.05) (Table 2). 

### 3.3. Validation of Top Differentially Expressed miRNAs

We conducted validation experiments for two randomly selected miRNAs with significantly higher expression in severe BPD (hsa-miR-628-5p, and hsa-miR-545*) and four miRNAs with significantly lower expression in severe BPD (hsa-miR-24-1*, hsa-miR-1255b, hsa-miR-205*, and hsa-let-7i*) by real-time PCR, using a subset of samples from mild/moderate BPD (*n* = 5) and severe BPD (*n* = 10) groups. Our results confirmed differential expression of all six miRNAs between the groups (*p* < 0.05) (Figure 4), validating the miRNA array results. 

### 3.4. Pathway Analysis

Ingenuity pathway analysis (IPA) of differentially expressed miRNAs in the two BPD groups indicated significant associations with molecular and cellular functions, including cell signaling, DNA replication, and cell cycle, as well as inflammatory response and disease, as summarized in Table 3. 

We also found that the top regulatory networks associated with the differentially expressed miRNAs included cell cycle; control of gene expression; cell signaling; and cellular function, assembly, and maintenance (Table 4). Of the 31 miRNAs identified in the array, 5 were associated with the most significant cellular and molecular networks found by IPA. These include three miRNAs with significantly lower expression in severe BPD when compared to mild/moderate BPD, namely miR-15* (mir-15a-3p), miR-615-3p, and miR-1255-5p, and two miRNAs with significantly higher expression in severe BPD, namely miR-185-5p (miR-185) and miR-9718 (which has the same seed sequence as miR-628-5p: UGCUGAC) (Table 4). A diagram of the identified networks, miRNA–gene target interactions, and connections among different networks based on experimental evidence is shown in Figure 5. The miRNA targets identified by IPA include cyclins, transcription factors, hormone receptors, and cell signaling enzymes (Table 4). 

Next, to expand our analysis beyond experimentally validated interactions, we used the IPA miRNA target filter to identify predicted miRNA–mRNA target relationships, based on miRNA seed sequence foreseen interactions. Among the top networks associated with predicted target genes, we identified organismal injury and respiratory disease pathways, as well as cellular trafficking, immunological disease, and cell and tissue development gene networks (Table 5). Graphical representations for the top associated networks are also shown in Appendix A.

## 4. Discussion

Many studies have focused on the prevention of severe BPD, but there is a paucity of literature on the biological basis of this disease’s pathogenesis. There is also a lack of validated biomarkers for BPD severity and targeted pharmacological therapies to improve the disease burden [37]. The current recommendations include optimizing ventilator settings while focusing on nutrition and addressing comorbidities [38]. This mostly stems from the fact that the exact underlying mechanisms of severe BPD are uncertain. In the current study, we sought to identify the molecular signatures associated with BPD severity to uncover regulatory pathways and new biomarkers for severe BPD. Through analysis of TAs, a noninvasive method of sample collection in mechanically ventilated infants, we discovered specific miRNA profiles in age-matched preterm newborns with mild/moderate and severe BPD. Our analysis revealed 31 differentially expressed miRNAs between subject groups. Of these, 4 miRNAs displayed at least 2-fold higher expression and FDR < 0.05 in severe BPD infants, and 27 miRNAs had significantly lower expression in severe BPD infants than in TAs from mild/moderate BPD infants. We also found associations of these miRNAs with functional gene networks related to lung inflammation, respiratory disease, oxidative stress, and cellular development, indicating that specific regulatory pathways may contribute to fundamental mechanisms of BPD severity.

Of the four miRNAs with higher expression in severe BPD infants, two (hsa-miR-545* and miR-185) have been previously implicated in mechanisms of cell cycle arrest, a known effect of prolonged mechanical ventilation and hyperoxia [39,40]. Specifically, miR-545 has been shown to suppress cell proliferation through targeting of cyclins and cyclin-related kinases [41], and miR-185 has been associated with induction of G1 cell cycle arrest and promotion of necroptosis and apoptosis via receptor-interacting kinases and caspase activity in alveolar epithelial type II cells [42,43]. Moreover, the role of mir-185 in hyperoxia-induced DNA damage and lung epithelial cell death triggered by oxidative stress has been reported using both animal models and human cells [43,44]. Importantly, extracellular vesicles containing miR-185 are significantly elevated following hyperoxia-induced cell death in alveolar type II epithelial cells [43]. While we did not characterize the composition of TAs regarding extracellular vesicles, it is possible that the increased levels of miR-185 found in TAs from severe BPD infants (who are subjected to higher and more prolonged hyperoxia than mild/moderate BPD infants) could represent a signal of hyperoxia-induced epithelial cell damage via increased extracellular vesicle miR-185 cargo. Therefore, our results could provide clinical evidence of a mechanism previously described in lung disease models. 

Two additional targets of miR-185 are the Rho GTPases, CDC42 and RHOA (Table 4), implicated in cell cycle, cellular function and maintenance, and cell signaling. By targeting these mRNAs, miR-185 has been shown to inhibit cell proliferation and migration in hepatic, colorectal, and lung cancer cell models [45,46,47,48]. Importantly, miR-185 can also interact with and target the long noncoding RNA FOXD2-AS1, known to inhibit cell proliferation, migration, and epithelial–mesenchymal transition (EMT) in glial cells and renal fibrosis models [49,50]. Regarding lung cells, Lei et al. demonstrated that miR-185 can reduce collagen V overexpression and EMT in pulmonary fibrosis [51]. While promotion of EMT in alveolar epithelial type II cells is known to alter normal alveolar development processes and contribute to BPD phenotypes, the effects of miR-185 in EMT processing during lung development and BPD progression have not been explored yet [52,53]. Finally, miR-185 has also been found to target the SOX9 gene and regulate Wnt signaling [54], a key regulator of lung organogenesis and development [55,56]. 

The remaining two miRNAs found overrepresented in TAs of severe BPD infants were miR-378 and miR-628-5p. With the exception of lung cancer [57], no studies to date have assessed the role of miR-378 in the lung. However, this small RNA has been implicated in several functions associated with BPD, including mitochondrial metabolism, autophagy, and oxidative stress responses in a variety of tissues [58,59]. On the other hand, miR-628-5p has been implicated in developmental functions such as embryonal implantation, as well as mechanisms of immune modulation, viral infection, and cell proliferation [60,61,62]. One study identified the fibroblast growth factor receptor 2 (FGFR2), the receptor for FGF10, as a target of miR-628-5p in ovarian cancer cells [63]. This gene has been identified as a key factor in early lung development [64], as well as alveolar epithelial type II cells homeostasis [65]. Interestingly, miR-628-5p has also been postulated as a potential biomarker for a variety of cancers in adults [66]. While no studies have yet assessed a direct role of this miRNA in lung development and disease, it could serve as a biomarker for severe BPD in neonates. 

Our analysis identified 27 miRNAs with low expression in severe BPD when compared to mild/moderate BPD. Of these, three were implicated in the top significant pathways identified by IPA (hsa-miR-15a*, hsa-miR-1255b-5p, and hsa-miR-615-3p). Specifically, these miRNAs were implicated in cell cycle and cell signaling functions, as well as disease processes in extrapulmonary tissues (Table 4). Regarding miR-15a, an avian model of long-term hypoxia stress revealed that its levels are influenced by hypoxia-induced factor 1 (HIF-1) during lung development [67]. Hypoxic episodes are common stressors associated with BPD secondary to immature respiratory center drive. Both acute and chronic hypoxic events associated with reduced peripheral chemosensitivity and proinflammatory responses occur during the course of neonatal BPD and are associated with poorer neurodevelopmental outcomes [68]. In addition, miR-15a has been shown to inhibit lung fibrosis by targeting the yes-associated protein 1 (YAP1), a key downstream effector of the Hippo pathway [69]. Therefore, it is possible that lower expression of this miRNA in severe BPD infants is associated with fibrotic phenotypes in severe BPD. It is worthwhile to mention that the Hippo pathway is a powerful regulator of lung development, cell differentiation, and tissue regeneration and homeostasis, and it has been found dysregulated in lymphocytes of patients with BPD [70,71]. Therefore, the role of miR-15a and the Hippo pathway in mechanisms leading to BPD severity warrants further investigation.

We detected and validated the downregulation of miR-1255b-5p in TAs of severe BPD infants. This miRNA was previously noted to regulate vascular endothelial growth factor A (VEGFA) expression in liver cirrhosis and hepatocellular carcinoma [72]. Abnormal VEGF and disrupted angiogenesis are described in lung tissues of infants that died of severe BPD [73]. In addition, miR-1255b-5p is downregulated in hypoxia-induced lung A549 cells [74], as well as in models of high altitude retinopathy [75], and in colorectal cancer, where it also regulates telomerase activity and suppresses EMT [76]. Notably, hypoxic stress in the presence of hyperoxia results in the alteration of alveologenesis in mouse models of neonatal BPD [77].

Through IPA analysis, we found that the downregulated miRNA miR-615-3p directly targets the ligand-dependent nuclear receptor corepressor (LCOR) and androgen receptor (AR) transcripts and indirectly targets the nuclear receptor peroxisome proliferator-activated receptor gamma (PPARG) (Table 4 and Figure 5). The AR is expressed in both male and female lung epithelial cells and can bind to DNA sequences to regulate the expression of cell differentiation genes [78,79]. Studies have also shown that the AR can mediate androgen-induced delays in fetal lung maturation [80,81,82]. For example, maternal treatment with dihydrotestosterone inhibits surfactant phospholipid production in the fetal lung, while treatment with the AR-selective antagonist flutamide enhances surfactant phospholipid production [83]. Thus, downregulation of miR-615-3p (a negative regulator of AR expression) in severe BPD could result in increased levels of AR, contributing to delays in lung maturation and surfactant expression. Similarly, PPARG contributes to homeostasis maintenance of epithelial–mesenchymal interactions, which are key players of lung organogenesis. Disruption of this process results in trans-differentiation of lung alveolar lipofibroblasts to myofibroblasts, contributing to the development of pulmonary fibrosis [84,85]. Importantly, PPARG agonists have been suggested as therapeutic targets for BPD due to their implications in the Wnt/β-catenin and TGF-β pathways [86]. Both pathways are induced by hypoxia, leading to decreased levels of PPARG and the subsequent differentiation of fibroblasts, leading to pulmonary fibrosis. Finally, because miR-615-3p is known to repress LCOR in macrophages [87], its downregulation in severe BPD may result in high LCOR levels, which also contribute to decreased PPARG. Overall, the miRNA signatures identified in severe BPD TAs can be associated with known mechanisms of lung inflammation, oxidative stress, developmental delay, and pulmonary fibrosis. More research is needed to explore their contributions to these mechanisms and their therapeutic potential.

The strengths of this study include the prospective nature of the design, using a cohort of preterm infants that is matched in age, sex, birth weight, method of delivery, and antenatal steroid exposure, and the identification of novel miRNA expression signatures associated with BPD phenotypes in TAs. An advantage of these samples is that they are readily available in intubated patients and may contain cellular makers that are very similar to those noted in the lungs. This study also excluded infants with pulmonary hypertension, which is a severe complication of BPD, hence providing more specificity to the signatures identified in the severe BPD cohort. 

We have also identified the following study limitations: First, this was a very small but representative sample of preterm neonates admitted to a single center, which affects the generalizability of the findings. It is also possible that with such a small sample size, the differential miRNA expression could be specific to the patients studied. However, the study design was exploratory in nature, utilizing convenience sampling in a small cohort of preterm infants receiving invasive mechanical ventilation. Second, given that the samples were collected at a single time point of the disease process, we are unable to determine whether the observed miRNA signatures are representative of the disease status or are only present at a specific time in the course of the pathology. This limits our ability to determine if there is a predictive potential to these markers or whether they represent a reflection of ongoing molecular pathways inherent to the disease severity or a trigger for molecular events related to disease phenotypes. Additionally, some key clinical variables were not fully explored in this study due to the small sample size. For example, the influence of sex differences, birth weight, fetal growth restriction, and exposure to antenatal corticosteroids on miRNA profile changes was not directly evaluated. Third, given that we studied preterm neonates of different developmental trajectories, it is possible that the differences we noted in the miRNA expression may reflect changes in cell populations present in the TA [88,89]. Lastly, it is becoming evident in recent studies that the microbiota in TAs can affect miRNA expression [90,91]. Because pulmonary inflammation secondary to perinatal and postnatal infection has been implicated as a key contributor in the pathogenesis of BPD, changes in the TA microbiome may be reflected in the observed miRNA signatures. Despite these limitations, this study demonstrates that miRNA profiling in TAs of preterm neonates can be used successfully as a potential biomarker for BPD severity. 

In summary, we have described miRNA expression signatures in TA samples obtained from extremely preterm infants with severe BPD and compared them to those of infants with mild to moderate BPD. We identified associations of these molecules with pathways involved in the disease phenotype, including lung inflammation, arrested cell division and development, and oxidative stress. We hypothesize that when identified in the early stages of the disease, these miRNAs can potentially predict the development of severe BPD and its associated phenotypes and hence could be a clinical tool for physicians to optimize ventilator settings, nutrition interventions, fluid balance, and aggressive treatment of infection. Future validation studies in larger cohorts of neonates at risk for severe BPD are needed to determine the effectiveness of these miRNA signatures as predictive and diagnostic tools for BPD severity, as well as to evaluate the significance of miRNA regulation in severe BPD progression.

## Figures and Tables

**Figure 1 biomedicines-09-00257-f001:**
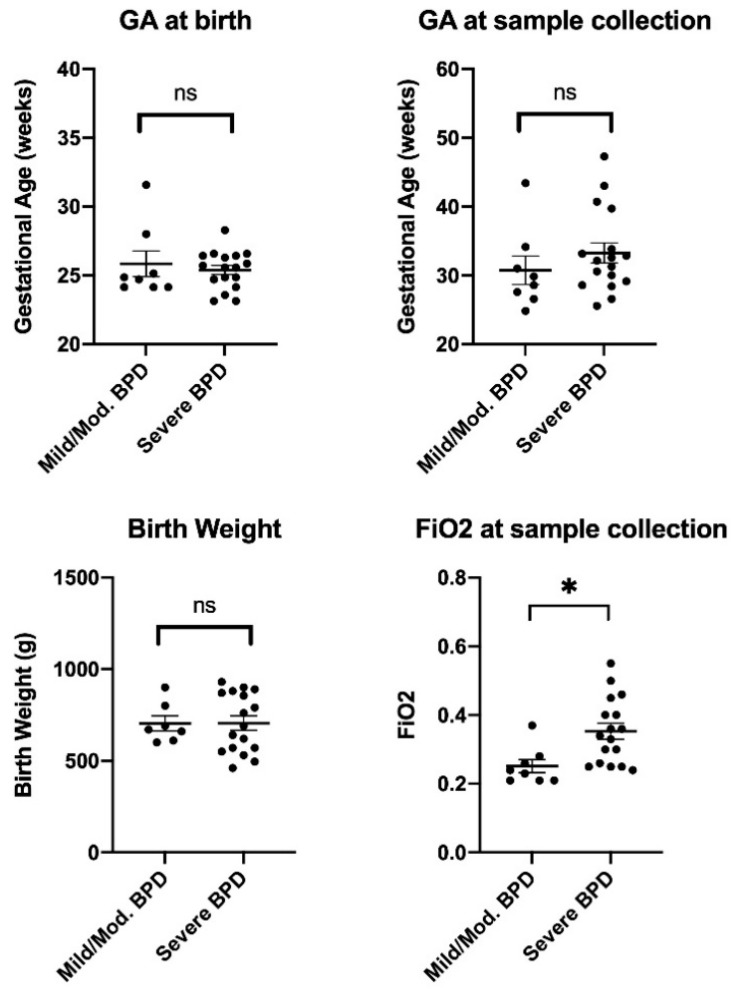
Patient demographics. Gestational age (GA) at birth and at sample collection time, birth weight, and fraction of inspired oxygen (FiO2) in mild/moderate (Mild/Mod.) and severe bronchopulmonary dysplasia (BPD) infants enrolled. No differences are observed between groups for GA or birth weight parameters. The FiO2 at sample collection time is significantly higher in the severe BPD group when compared to the Mild/Mod. BPD group (unpaired *t*-test, * *p* = 0.0104).

**Figure 2 biomedicines-09-00257-f002:**
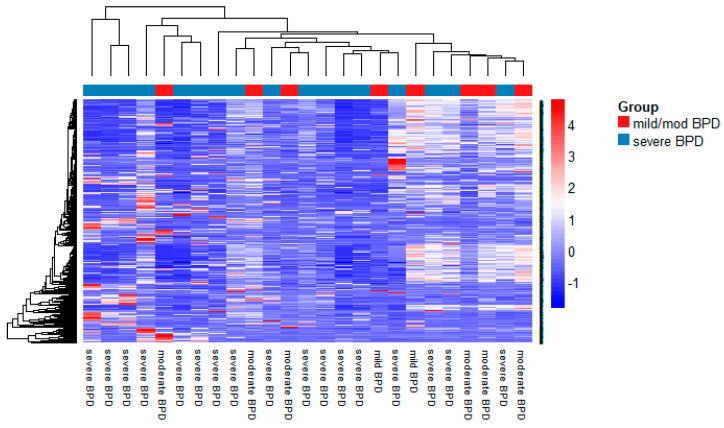
Heatmap of expression of 1048 miRNAs in tracheal aspirates from mild/moderate (*n* = 8) and severe (*n* = 17) BPD infants (normalized by global mean), obtained by PCR arrays. Heatmap generated in R using the NMF package as described in Section 2.

**Figure 3 biomedicines-09-00257-f003:**
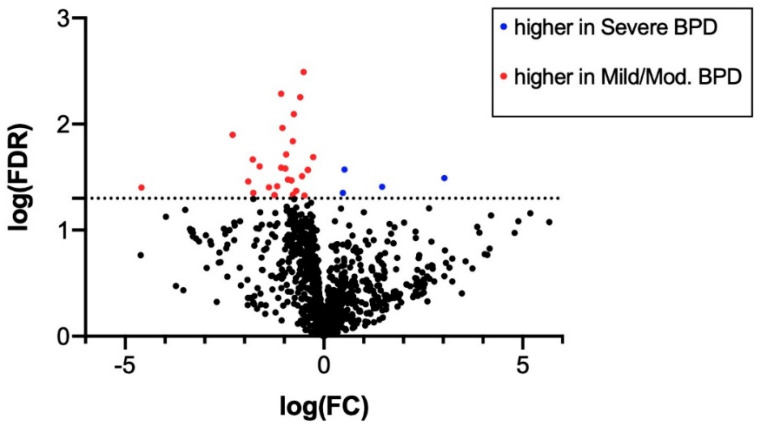
Volcano plot of miRNAs expressed in tracheal aspirates from mild/moderate (*n* = 8) and severe (*n* = 17) BPD infants. Red dots indicate miRNAs upregulated in mild/moderate BPD samples. Blue dots indicate miRNAs with higher expression in severe BPD samples. FC: fold change; FDR: false discovery rate. Dotted line indicates log(FDR) = 1.3, corresponding to FDR = 0.05. Volcano plot was generated in GraphPad Prism software as described in Section 2.

**Figure 4 biomedicines-09-00257-f004:**
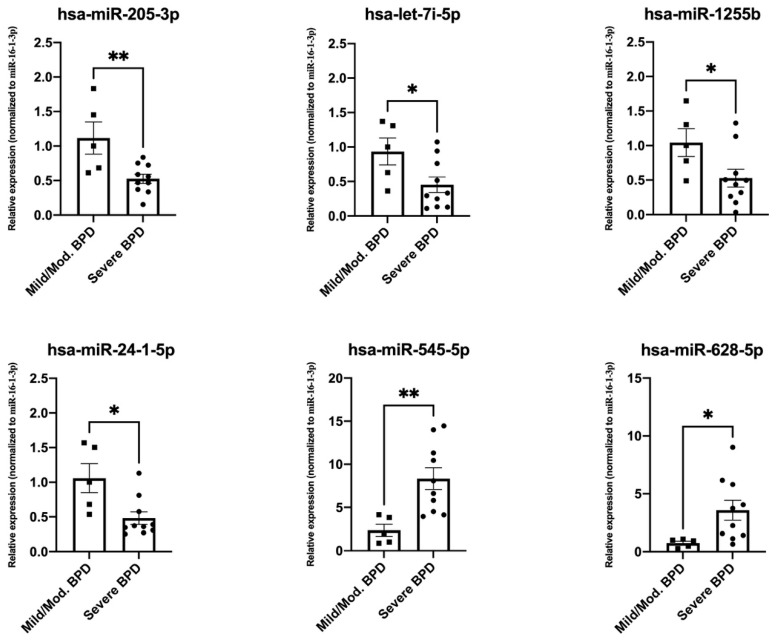
Validation experiments by real-time PCR and specific primers for hsa-miR-205-3p, hsa-let-7i-5p, hsa-miR-1255b, hsa-miR-24-1-5p, hsa-miR-545-5p, and hsa-miR-628-5p in a subset of mild/moderate (Mild/Mod.) BPD samples (*n* = 5) and severe BPD samples (*n* = 10). Y axes indicate relative expression values after normalization to hsa-miR-16-1-3p, and a mild/moderate calibration sample using the 2^−ΔΔ*C*T^ method [32,35]. Significant differences were determined by t-test using the GraphPad software (* *p* < 0.05, ** *p* < 0.01).

**Figure 5 biomedicines-09-00257-f005:**
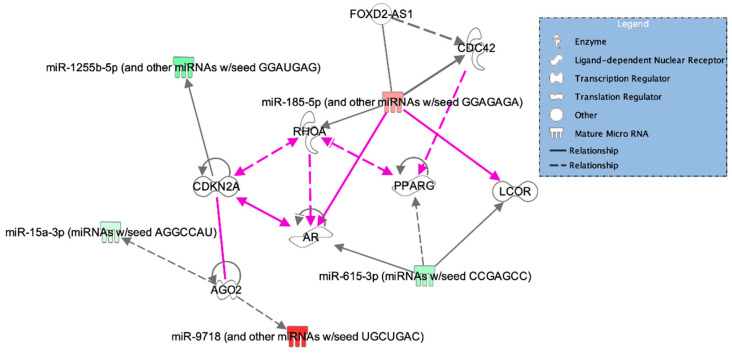
Graphical representation for the top 4 networks associated with differentially expressed miRNAs in severe BPD samples. Grey lines represent individual pathways, and pink lines indicate interactions among pathways. Full and dotted lines indicate direct and indirect relationships between molecules, respectively. miRNAs with higher expression in the severe BPD group are shown in shades of red, according to upregulation intensity. Similarly, miRNAs with lower expression in severe BPD are shown in shades of green. AGO2: argonaut RISC (RNA-induced silencing complex) catalytic component 2; CDKN2A: cyclin-dependent kinase inhibitor 2A; AR: androgen receptor; LCOR: ligand-dependent nuclear receptor corepressor; CDC42: cell division cycle 42; FOXD2-AS1: FOXD2 (forkhead box D2) adjacent opposite strand RNA 1; PPARG: peroxisome proliferator-activated receptor gamma; RHOA: Ras homolog family member A.

**Table 1 biomedicines-09-00257-t001:** Patient demographics at study enrollment.

Characteristic	Mild/Moderate BPD (*n* = 8)	Severe BPD (*n* = 17)	*p*-Value
Gestational age (GA) at birth, weeks (mean ± SD)	25.84 ± 2.64	25.39 ± 1.38	0.584
GA at sample collection, weeks (mean ± SD)	30.75 ± 5.86	33.26 ± 6.02	0.337
Birth weight, grams (mean ± SD)	704 ± 108	706 ± 162	0.981
Day of life (mean ± SD)	37 ± 31	60 ± 40	0.182
Male sex, % (*n*)	50 (4)	47 (8)	>0.999
FiO_2_ at sample collection (mean ± SD)	0.25 ± 0.05	0.35 ± 0.10	**0.010**
Antenatal steroids exposure, % (*n*) ^1^	75 (6)	76 (13)	0.289
Delivered via C-section, % (*n*)	75 (6)	88 (15)	0.289
**Racial/Ethnic Group, % (*n*)**			
Non-Hispanic White	63 (5)	53 (9)	0.508
Non-Hispanic Black	25 (2)	6 (1)	0.289
Hispanic	0 (0)	18 (3)	**0.008**
Asian	13 (1)	18 (3)	0.070
More than one race	0 (0)	6 (1)	**0.008**

^1^ Any antenatal steroids exposure (1 or more doses received by mother before birth), values in bold indicate *p* = 0.05.

**Table 2 biomedicines-09-00257-t002:** Differentially expressed miRNAs in severe vs. mild/moderate BPD tracheal aspirates (Tas).

miRNA ID	log(Fold Change)	Average Expression	FDR
hsa-miR-628-5p	3.031	2.823	0.032
hsa-miR-185	1.463	1.591	0.039
hsa-miR-545*	0.516	0.361	0.027
hsa-miR-378	0.479	0.607	0.044
hsa-miR-3713	−0.270	0.152	0.020
hsa-miR-3151	−0.404	0.414	0.027
hsa-miR-1295	−0.404	0.495	0.027
hsa-miR-1286	−0.488	0.487	0.047
hsa-miR-380	−0.509	0.318	0.003
hsa-miR-15a*	−0.550	0.851	0.031
hsa-miR-3175	−0.599	0.427	0.006
hsa-miR-493	−0.698	1.030	0.042
hsa-miR-3193	−0.759	0.537	0.008
hsa-miR-105*	−0.781	0.879	0.014
hsa-miR-4300	−0.784	1.019	0.046
hsa-miR-631	−0.815	0.756	0.034
hsa-miR-2116*	−0.902	0.954	0.033
hsa-miR-4304	−0.951	0.986	0.019
hsa-miR-3125	−0.971	1.316	0.026
hsa-miR-4303	−1.046	1.158	0.011
hsa-miR-1908	−1.078	0.960	0.005
hsa-miR-205*	−1.079	1.376	0.026
hsa-miR-3674	−1.176	1.101	0.039
hsa-miR-615-3p	−1.243	1.649	0.047
hsa-miR-4305	−1.383	1.613	0.040
hsa-let-7i*	−1.615	1.969	0.025
hsa-miR-4330	−1.776	1.918	0.045
hsa-miR-1255b	−1.787	1.733	0.022
hsa-miR-125b-1*	−1.905	2.199	0.035
hsa-miR-24-1*	−2.300	1.153	0.013
hsa-miR-646	−4.590	5.529	0.040

**Table 3 biomedicines-09-00257-t003:** IPA pathways associated with differentially expressed miRNAs.

Top Pathways	*p*-Value
**Top molecular and cellular functions**	
Cell-to-cell signaling and interaction	9.63 × 10^−4^–9.63 × 10^−4^
Cellular function and maintenance	9.63 × 10^−4^–9.63 × 10^−4^
DNA replication, recombination, and repair	1.39 × 10^−2^–1.39 × 10^−2^
Cell death and survival	1.86 × 10^−2^–1.86 × 10^−2^
Cell cycle	2.57 × 10^−2^–2.57 × 10^−2^
**Top diseases and disorders**	
Inflammatory disease	2.08 × 10^−5^–2.08 × 10^−5^
Inflammatory response	1.89 × 10^−3^–2.08 × 10^−5^
Organismal injury and abnormalities	3.97 × 10^−2^–2.08 × 10^−5^
Renal and urological disease	2.08 × 10^−5^–2.08 × 10^−5^
Reproductive system disease	3.97 × 10^−2^–2.01 × 10^−4^

**Table 4 biomedicines-09-00257-t004:** IPA core analysis top associated networks and molecules (miRNAs and target genes).

Top Associated Networks	Molecules	Score
Cell cycle, gene expression, RNA post-transcriptional modification	AGO2, miR-15a-3p, miR-9718	6
Cell-to-cell signaling and interaction, cellular assembly and organization, small molecule biochemistry	AR, LCOR, miR-615-3p, PPARG	3
Cell cycle, cellular function and maintenance, cell signaling	CDC42, FOXD2, RHOA, miR-185-5p	3
Cardiovascular disease, cardiovascular system development and function, cancer	CDKN2A, miR-1255b-5p	3

AGO2: argonaut RISC (RNA-induced silencing complex) catalytic component 2; AR: androgen receptor; LCOR: ligand-dependent nuclear receptor corepressor; PPARG: peroxisome proliferator-activated receptor gamma; CDC42: cell division cycle 42; FOXD2-AS1: FOXD2 (forkhead box D2) adjacent opposite strand RNA 1; RHOA: Ras homolog family member A; CDKN2A: cyclin-dependent kinase inhibitor 2A.

**Table 5 biomedicines-09-00257-t005:** IPA top associated networks based on miRNA–mRNA binding relationships.

Top Associated Networks	Score
Respiratory disease, cancer, organismal injury and abnormalities	37
Cellular movement, immune cell trafficking, immunological disease	35
Cell-to-cell signaling and interaction, cellular assembly and organization, tissue development	33
Cell-to-cell signaling and interaction, cellular development, cellular growth and proliferation	29
Cellular movement, hematological system development and function, immune cell trafficking	27

## Data Availability

The data presented in this study are openly available in Gene Expression Omnibus at https://www.ncbi.nlm.nih.gov/geo/query/acc.cgi?acc=GSE165828. The code used for data analysis in the current study can be found at the Silveyra lab repository, http://psilveyra.github.io/silveyralab/.

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
