# Peer review of "MicroRNA Signatures Associated with Bronchopulmonary Dysplasia Severity in Tracheal Aspirates of Preterm Infants"

_biomedicines, 2021, doi:10.3390/biomedicines9030257_

Round 1

Reviewer 1 Report

The results of miRNA research in this manuscript are very interesting. The results provide important information to the readers, especially who work in the lung research field. The authors described and discussed the miRNA signatures they found significant differences between the mild/moderate BPD and severe BPD, but the authors did not discuss how those miRNAs can contribute as biomarkers. The authors wrote in the end of the Abstract, “These signatures may serve as biomarkers of disease severity in infants with BPD.” I think the readers would take interest how those miRNAs can be biomarkers for predicting severity. It would be better if that kind of discussion included in the Discussion.

Author Response

We thank the reviewer for this suggestion. Following the reviewer’s recommendation, we have incorporated additional information on the potential use of miRNAs as biomarkers for BPD severity in neonates in the last paragraph of the revised discussion.

Reviewer 2 Report

Review

Journal: biomedicines

Article: Tracheal aspirate miRNA signatures in preterm infants with severe bronchopulmonary dysplasia

The current manuscript aims to analyze the expression and profile of miRNAs present in TAs obtained from a cohort of neonates with mild/moderate and severe BPD, identifying patients at risk.

In general, the study is well conducted, the topic is of extreme importance, and the authors put their findings in the context of the current literature. However, inflammation is a important factor that is associated and observed in the majority of the cases developing BPD. Therefore, this should be already mentioned in the abstract.

Furthermore, the impact of maternal health on chronic inflammation is not mentioned. This would be worth including into the 2nd paragraph of the introduction.

Since the authors investigate the association of miRNA and BPD a paragraph including the current knowledge on this should be included. miRNA 29b and 219-5p are associated with BPD

There are various pathways involved in the development of BPD that have been investigated in animal models including cyclooxygenase-2 in mice that should be mentioned.

Author Response

We thank the reviewer for his/her comments. Following the reviewer’s suggestions, we have incorporated the contributions of inflammation to BPD development in the revised introduction and abstract. We have also expanded the description of BPD etiology incorporating the contributions of maternal health in the second part of the introduction. Regarding previously identified miRNAs involved in BPD pathogenesis and associated pathways, we have now included a summary of recent work conducted on these topics in the fourth paragraph of the revised introduction. To clarify that our study focuses on BPD severity (mild/moderate vs. severe) rather than development, we have also now edited the title of the study.